# Persistent high mortality rates for Diabetes Mellitus and Hypertension after excluding deaths associated with COVID-19 in Brazil, 2020–2022

**Rodrigo Moreira**[1], **Leonardo S. Bastos**[2], **Luiz Max Carvalho**[3], **Laís Picinini Freitas**[4,5], **Antonio G. Pacheco**[2]*

1 Instituto Nacional de Infectologia Evandro Chagas/Fiocruz, Rio de Janeiro, Brasil, 2 Programa de Computação Científica, Fundação Oswaldo Cruz, Rio de Janeiro, Brasil, 3 Escola de Matemática Aplicada, Fundação Getulio Vargas, Rio de Janeiro, Brasil, 4 École de Santé Publique, Université de Montréal, Montreal, Canada, 5 Centre de Recherche en Santé Publique, Montreal, Canada

* antgui@gmail.com

**Data Availability Statement:** The mortality data used in this study is sourced from the Brazilian mortality system (SIM - Sistema de Informação

## Abstract

### Introduction

The outbreak of severe acute respiratory syndrome coronavirus 2 (SARS-CoV-2) posed a significant public health challenge globally, with Brazil being no exception. Excess mortality during this period reached alarming levels. Cardiovascular diseases (CVD), Systemic Hypertension (HTN), and Diabetes Mellitus (DM) were associated with increased mortality. However, the specific impact of DM and HTN on mortality during the pandemic remains poorly understood.

### Methods

This study analyzed mortality data from Brazil's mortality system, covering the period from 2015 to 2022. Data included all causes of death as listed on death certificates, categorized by International Classification of Diseases 10th edition (ICD-10) codes. Population data were obtained from the Brazilian Census. Mortality ratios (MRs) were calculated by comparing death rates in 2020, 2021, and 2022 to the average rates from 2015 to 2019. Adjusted MRs were calculated using Poisson models.

### Results

Between 2015 and 2022, Brazil recorded a total of 11,423,288 deaths. Death rates remained relatively stable until 2019 but experienced a sharp increase in 2020 and 2021. In 2022, although a decrease was observed, it did not return to pre-pandemic levels. This trend persisted even when analyzing records mentioning DM, HTN, or CVD. Excluding death certificates mentioning COVID-19 codes, the trends still showed increases from 2020 through 2022, though less pronounced.

sobre Mortalidade), which is publicly available from DATASUS (Opendatasus - https://opendatasus.saude.gov.br/dataset/sim). Additionally, population data were obtained from the Brazilian Census Bureau (IBGE – Instituto Brasileiro de Geografia e Estatística) through the SIDRA system (http://api.sidra.ibge.gov.br/).

**Funding:** This work was supported by Fundação de Amparo a Pesquisa do Estado do Rio de Janeiro (FAPERJ) [E-26/203.172/2017 to A.G.P.] and Conselho Nacional de Desenvolvimento Científico e Tecnológico (CNPq) [310566/2021-5 to A.G.P.]. The funders had no role in study design, data collection and analysis, decision to publish, or preparation of the manuscript.

**Competing interests:** The authors have declared that no competing interests exist.

## Conclusion

This study highlights the persistent high mortality rates for DM and HTN in Brazil during the years 2020–2022, even after excluding deaths associated with COVID-19. These findings emphasize the need for continued attention to managing and preventing DM and HTN as part of public health strategies, both during and beyond the COVID-19 pandemic. There are complex interactions between these conditions and the pandemic's impact on mortality rates.

## Introduction

Severe acute respiratory syndrome coronavirus 2 (SARS-CoV-2) disease (COVID-19) has been a major public health emergency worldwide and in Brazil [1] with high burden, hitting the hardest in 2020 and 2021. Excess mortality during that period reached very high values in many countries. One study pointed to excess rates as high as 734.9 per 100,000 inhabitants in Bolivia [2] In Brazil, even though excess mortality was heterogeneous among states of residence, a rate of 186.9 per 100.000 was reported in that same study. Other studies reported excess deaths ranging from 10% to 40% in that same period in Brazil [3–5].

Cardiovascular diseases (CVD) and associated conditions such as Systemic Hypertension (HTN) and Diabetes Mellitus (DM) have been associated with severe COVID-19 and mortality [1, 6, 7]. However, the role of HTN and DM as independent risk factors are not yet clear [8]. Prior to the pandemic, CVD, DM, and HTN were already major contributors to substantial morbidity and mortality implications in the general population. However, the advent of COVID-19 has further exacerbated these challenges. People with chronic conditions such as DM, CVD and HTN are at increased risk of hospitalization and mortality in SARS-CoV [9]. At the peak of the pandemic as many as 50% of patients reported having at least one of these comorbidities upon being hospitalized with COVID-19 [10]. Several meta-analyses from observational studies have reported relative risks for death in patients with COVID-19 from 1.5 to 2.0 for DM and HTN) [11–13] the main contributors of CVD risk. Although the exact mechanisms underlying this increased mortality are not fully understood, insights from epidemiological studies provide some evidence. Chronic hyperglycemia and the associated chronic inflammatory state in DM can significantly compromise the body's immune function and increase the risk of complications related to infection and inflammation [14]. Moreover, the viral invasion pathway of SARS-CoV-2 seems complex. The virus initiates human infection binding to the angiotensin-converting enzyme 2 (ACE2) receptor on the cell surface. ACE2, is a key component of the renin-angiotensin system and a target of antihypertensive medications [10]. This interaction between COVID-19 and ACE2 may potentially contribute to worse clinical outcomes in hypertensive patients.

While excess mortality and increased rates of CVD mortality have been reported worldwide and in Brazil [15, 16], robust evidence on DM and HTN as contributing morbidities during the pandemic is scarce.

The present study aims to address this gap by analyzing public health data at the national level to investigate the contributing role of CVD, DM, and HTN on mortality impact in Brazil, both before (2015–2019) and during the COVID-19 pandemic (2020–2022) in Brazil.

## Methods

In this study we compared sex, age and state of residence adjusted mortality ratios (aMR) in Brazil in 2020–2022, compared to the preceding period of 2015–2019 and in subgroups of

CVD, DM and HTN whenever these conditions were mentioned on death certificates. Comparisons were made with and without COVID-19 mentioned on the death certificates.

Mortality data used in this study comes from the Brazilian mortality system (SIM—*Sistema de Informação sobre Mortalidade*) and is publicly available from DATASUS (Opendatasus - https://opendatasus.saude.gov.br/dataset/sim). Files for all-cause mortality, including multiple causes as assigned on death certificates, from January 2015 through December 2022, were downloaded and processed as described below. The 2022 database was deemed as preliminary data at the time of this analysis (accessed on Aug/30/2023).

All data from SIM consisted of variables extracted from a digital version of death certificates except information that could identify individuals and were available for analysis, including all causes of death (CoD) mentioned on death certificates and the underlying cause of death, which is a calculated variable, based on the information of immediate, contributing and concomitant causes leading to death. All causes are coded into International Classification of Diseases 10th. edition (ICD-10) codes and can thus be grouped according to what is being studied.

Population data were obtained from the Brazilian Census Bureau (IBGE–*Instituto Brasileiro de Geografia e Estatística*) through the SIDRA system(http://api.sidra.ibge.gov.br/). We obtained population projections per age group, sex and state of residence from January 2015 to December 2022.

For this study, we worked with ICD-10 codes of interest mentioned in any field where those codes are expected to be found (i.e. section VI-49). The following groups were created:

- COVID-19

- ICDs: B342, U071, U072

- Diabetes Mellitus (DM)

- ICDs: E10 through E14

- High blood pressure (HBP)

- ICDs: I10 through I15

- CVD: Cardiovascular diseases

- ICDs: I00 through I99, except I46 (cardiac arrest)

Mortality ratios (MRs) were calculated as the death rates in 2020, 2021 and 2022 over the average death rates from 2015 through 2019. Values above one were considered excess when compared to the non-pandemic period. The numerator of the rates included all death certificates that mentioned any ICD-10 codes described above in any field of the death certificate. To measure the impact of COVID-19, we also calculated the mortality rate excluding death certificates that mentioned COVID-19 in any field.

Adjusted MRs were calculated through Poisson models using the log of the population as an offset. All models were adjusted for state of residence, age group and sex. When controlling for state of residence, mixed-effect models with random intercepts were used.

### Ethical statement

Our research uses publicly available Information and aggregated data without individual identification (DataSUS and IBGE data). Thus, an exemption from submission to the Institutional Review Board is provided by federal Brazilian resolution, CNS n.˚ 510, from 2016.

All analyses were performed with R 4.2.2 [17].

## Results

A total of 11,423,288 deaths were recorded in Brazil from 2015 to 2022. The observed mortality rates were fairly stable over time until 2019, after which there was a steep rise in 2020 and 2021. In 2022 there was a decrease, but not to the same level as before (Fig 1).

This pattern was also noticed when only records that mentioned DM, HTN or CVD were selected. When records that also mentioned COVID-19 codes were excluded, all trends still presented increases from 2020 through 2022, though much smoother (Fig 1).

Overall characteristics of deaths in Brazil are depicted in S1 Table.

Overall, adjusted mortality ratios were 9% and 24% higher for 2020 and 2021, respectively, compared to 2015–2019, whereas for 2022 it was 2% lower. For all three years, when COVID-

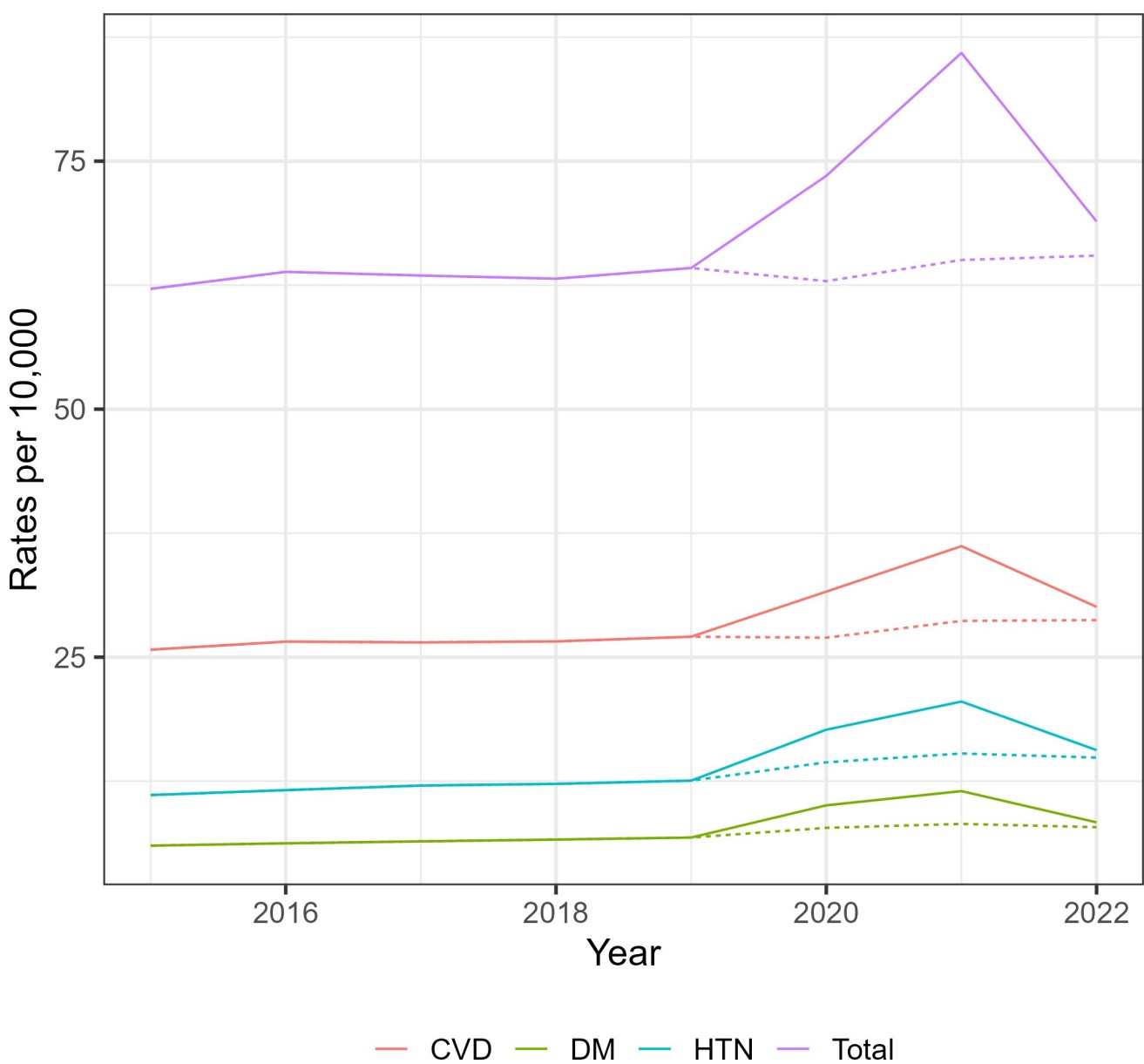

**Fig 1. Mortality rates in Brazil with (solid lines) and without (dotted lines) COVID-19 mentioned in the death certificate—overall, cardiovascular disease (CVD), Diabetes Mellitus (DM) and Hypertension (HTN) - 2015–2022.**

 

**Table 1. Adjusted mortality ratios (95% confidence intervals) comparisons among individuals who had Diabetes Mellitus (DM), Hypertension (HTN) or cardiovascular disease (CVD) mentioned in their death certificate with and without mention of COVID-19.**

| Category | 2020 | 2021 | 2022 |
|---|---|---|---|
| Total | 1.087 (1.085–1.089) | 1.243 (1.241–1.245) | 0.976 (0.974–0.977) |
| Total, no COVID | 0.93 (0.929–0.932) | 0.941 (0.939–0.943) | 0.927 (0.926–0.929) |
| DM | 1.446 (1.439–1.453) | 1.607 (1.6–1.615) | 1.136 (1.13–1.142) |
| DM, no COVID | 1.119 (1.113–1.125) | 1.145 (1.139–1.151) | 1.069 (1.063–1.075) |
| HTN | 1.367 (1.362–1.372) | 1.543 (1.538–1.548) | 1.143 (1.139–1.147) |
| HTN, no COVID | 1.113 (1.108–1.117) | 1.149 (1.145–1.154) | 1.087 (1.083–1.092) |
| CVD | 1.102 (1.099–1.105) | 1.228 (1.225–1.231) | 0.994 (0.991–0.997) |
| CVD, no COVID | 0.94 (0.937–0.942) | 0.972 (0.97–0.975) | 0.95 (0.947–0.952) |

19 was removed from the numerator, total aMRs fell back to values around 0.93. A similar configuration is noted for CVD mortality rates (Table 1).

When we look at mortality rates for DM and HTN, aMRs are higher than those for overall and CVD (ranging from 1.14 in 2022 to 1.61 in 2021 for DM and 1.14 in 2022 to 1.54 in 2021), but they did not fall below 1 when COVID-19 records are excluded, reaching a 15% increase for both conditions in 2021 (Table 1).

Those figures were not homogeneous when we look within age subgroups either. As shown in Fig 2.

Significant increased ratios were noted from 30 years and older for all groups in 2020 and 2021 but not in 2022, when looking at all causes of death (Panel A). In 2021 middle-aged adults were particularly impacted with a 48% increase in overall mortality in the 40–49-year-old group. All ratios return to less than 1 when COVID-19 is removed. A similar pattern was noted for CVD (Panel D). For DM (Panel B) and HTN (Panel C), not only does the increase in the ratios begin in younger age groups (0–19 and 20–29, respectively), but also they did not return to baseline values when death certificates that mentioned COVID-19 were removed from the analysis. Similar patterns are noted for sex (S1 Fig). Across states, figures were heterogeneous. In 2020 overall mortality rates increased for most states (S2 Fig), ranging from -2% in Rio Grande do Sul to 30% increase in Amazonas, whereas in 2021 it ranged from 12% (Alagoas and Sergipe) up to 49% in Rondônia. Of note, Amazonas, one of the states with the deadliest outcomes during the pandemic experienced a 47% increase rate in 2021 on top of that 30% in 2020. All ratios also returned to baseline values upon the removal of COVID-19. In 2022 some states did have significantly increased ratios, but all of them returned to values below 1 when COVID-19 was removed. Similar patterns were observed for CVD (S5 Fig). Regarding DM and HTN (S3 and S4 Figs) it followed the same trends as the overall mortality ratios, once again with AM displaying higher aMR among all the states.

## Discussion

In this study, we showed increased mortality rates during the COVID-19 pandemic in Brazil among deceased individuals who had CVD, HTN and DM mentioned in their death certificates. Even though those are expected results, rates did not return to baseline values after removing cases with concomitant COVID-19 for DM and HTN, as happened with overall (total) and the CVD subgroup.

Our results are in line with the literature, in terms of the overall and CVD mortality ratios. In Brazil, overall excess mortality ranged from 10% to 40%, depending on the study and period studied [3, 5], which is close to our mortality ratios in 2020 (9%) and 2021 (24%). If we did

 

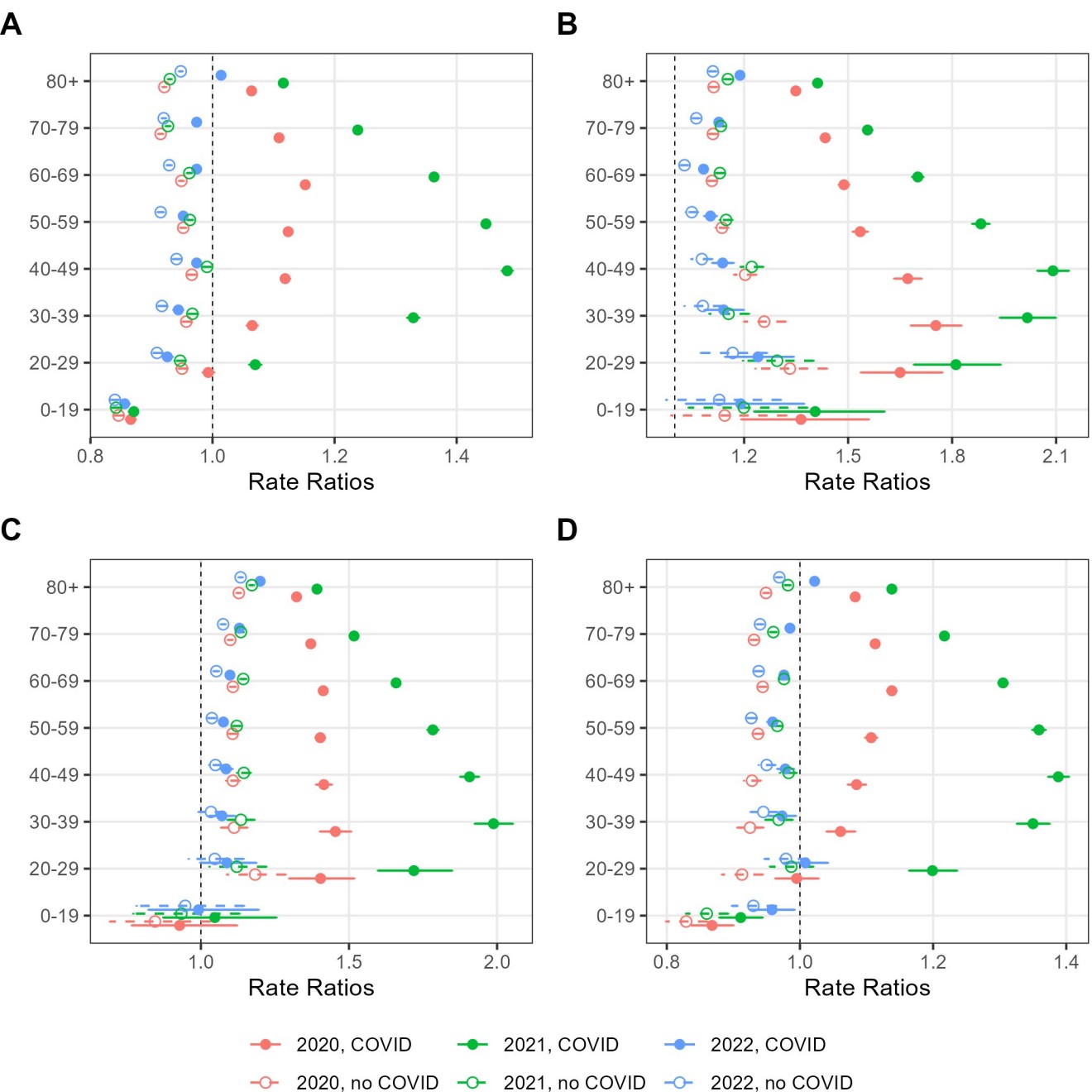

**Fig 2. Adjusted mortality ratios in Brazil with and without COVID-19 mentioned in the death certificate per age group.** A—Overall, B–DM, C–HTN, D–CVD; baseline—2015–2019.

look at excess mortality, our data would show figures even closer to those (14% and 32.4%, respectively). CVD mortality rates were about the same level as overall rates and are also in line with the literature [15].

It is worth highlighting that Brazil faced many challenges in dealing with COVID-19, at some point having the highest number of cases and deaths in Latin America [18] Among the regions most impacted was the state of Amazonas which experienced the highest mortality rates in the country during the pandemic. Known disparities in healthcare access,

overwhelming of the healthcare system and quality of care during this period, leading to delayed or inadequate treatment for COVID-19 might have contributed to such results.

We showed higher mortality rates from CVD in the presence of COVID-19 in this study. Viral infection can initiate myocardial injury and provoke inflammatory hyperactivity. Among the most prevalent cardiovascular complications observed in COVID-19 are myocardial infarction, myocarditis accompanied by reduced systolic function of the left ventricle, arrhythmias and thromboembolic complications [19].

As expected, death rates for those with HTN and DM as comorbidities also significantly increased in 2020 and 2021, but contrary to overall deaths and CVD, they did not return to baseline values after death certificates that mentioned COVID-19 were removed. Two different mechanisms may be playing a role in this pattern. The first one is the possibility of underreporting of COVID-19 among deceased individuals with those conditions in Brazil, during that period. Even though this mechanism is probably present in this case, and has been reported in Brazil [20], we do not believe this is the only factor playing a role here. First because overall and CVD death rates returned to baseline, and even to values lower than those in 2015–2019, since it is expected that mortality by causes not related to COVID-19 would decrease [20], and, because we removed all death certificates that mentioned COVID-19 and not only those with COVID-19 as the underlying cause of death, which would account for cases where misclassification occurred.

The other mechanism would be that these conditions had their prevalence increased during the pandemic, and contributed to increased death rates during this period, and even in 2022, as shown. The relationship of DM and HTN with life-style changes during the pandemic and even after it (e.g. decreased physical activity, poor glycemic control and adherence to therapy) may have had an impact on aMR in the population without COVID-19 [21]. Moreover, some studies pointed to newly diagnosed HTN and DM after acute COVID-19 diagnosis, especially in severe cases [22, 23].

One possibility both for risk factors and increased prevalence would be the fact that most individuals with those conditions are older and would be driving this trend, as their death rates are higher than younger individuals [8]. Our results were adjusted for age, sex and state of residence, which should limit the impact of those distributions across groups. Moreover, we showed that rates in all age groups above 20–29 years have the same behavior, with a predominance among young adults (Fig 2).

This study has several limitations. First, it is based on data from death certificates, which are known to suffer from misclassification, despite going through thorough revision before the information is entered into the database [24]. This problem results in both underreporting of conditions (especially when garbage ICD-10 codes are reported as an underlying cause of death) or reporting wrong diagnoses. The approach used in this study to search for all causes mentioned tends to lower this problem and has been used before to study causes of death in Brazil [25]. Moreover, census data for the year 2022 were not available at the time of this study. Therefore, population estimates for 2022 were based on official projections (IBGE) from 2015 to December 2022. These projections may introduce uncertainty into the calculation of mortality rates and ratios, particularly if population dynamics deviate significantly from the projected trends.

Another limitation is the number of variables used to adjust the rates. Even though race/ethnicity, education and other characteristics are also available on death certificates, there are difficulties with respect to missing values and population projection for those subgroups. Since in our case we wanted to control for major factors that would impact HTN and DM rates, we believe that sex, age and state of residence would cover most heterogeneities involved in those calculations.

Strong points of our study include the use of a large, representative database in Brazil, that comprises all deaths reported to the Ministry of Health and the possibility to study multiple causes of death instead of the underlying causes, as is usually done in mortality studies.

In conclusion, our study showed persistent higher mortality in individuals with a diagnosis of HTN and DM at the time of death in Brazil during the COVID-19 pandemic, even after removing those deaths related to COVID-19. This finding should point to improving the diagnosis of COVID-19 and correctly reporting it on the death certificate, and also increase surveillance for both HTN and DM in patients who recently had a COVID-19 diagnosis to better control those conditions among them.

## Supporting information

**S1 Fig.** Adjusted Mortality Ratios in Brazil with and without COVID-19 mentioned in the death certificate per age group–A—Overall, B–DM, C–HTN, D–CVD; baseline—2015–2019). (TIFF)

**S2 Fig. State of residence.**
(TIFF)

**S3 Fig. State of residence.**
(TIFF)

**S4 Fig. State of residence.**
(TIFF)

**S5 Fig. State of residence.**
(TIFF)

**S1 Table. Characteristics of mortality data in Brazil, 2015–2022.**
(DOCX)

## Author Contributions

**Conceptualization:** Leonardo S. Bastos, Luiz Max Carvalho, Laís Picinini Freitas, Antonio G. Pacheco.

**Data curation:** Antonio G. Pacheco.

**Formal analysis:** Antonio G. Pacheco.

**Methodology:** Antonio G. Pacheco.

**Writing – original draft:** Rodrigo Moreira, Leonardo S. Bastos, Luiz Max Carvalho, Laís Picinini Freitas, Antonio G. Pacheco.

**Writing – review & editing:** Rodrigo Moreira, Leonardo S. Bastos, Luiz Max Carvalho, Laís Picinini Freitas, Antonio G. Pacheco.

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
