## [Decision Letter · Decision Letter 0]

10 Jan 2024

PGPH-D-23-02038

Persistent high mortality rates for Diabetes Mellitus and Hypertension after excluding deaths associated with COVID-19 in Brazil, 2020-2022

Dear Dr. Moreira,

Thank you for submitting your manuscript to PLOS Global Public Health. After careful consideration, we feel that it has merit but does not fully meet PLOS Global Public Health’s publication criteria as it currently stands. Therefore, we invite you to submit a revised version of the manuscript that addresses the points raised during the review process.

The manuscript has been evaluated by two reviewers, and their comments are available below.

The reviewers have raised a number of concerns and request clarification of various aspects of the methods and analyses.

Could you please carefully revise the manuscript to address all comments raised?

We look forward to receiving your revised manuscript.

Kind regards,

Steve Zimmerman, PhD

PLOS Staff Editor

Journal Requirements:

1. Please provide separate figure files in .tif or .eps format only and remove any figures embedded in your manuscript file. Please also ensure all files are under our size limit of 10MB.

2. In the online submission form, you indicated that "Data will be available upon request". All PLOS journals now require all data underlying the findings described in their manuscript to be freely available to other researchers, either 1. In a public repository, 2. Within the manuscript itself, or 3. Uploaded as supplementary information.

Additional Editor Comments (if provided):

Reviewers' comments:

Reviewer's Responses to Questions

**Comments to the Author**

1. Does this manuscript meet PLOS Global Public Health’s publication criteria? Is the manuscript technically sound, and do the data support the conclusions? The manuscript must describe methodologically and ethically rigorous research with conclusions that are appropriately drawn based on the data presented.

Reviewer #1: Yes

Reviewer #2: Yes

2. Has the statistical analysis been performed appropriately and rigorously?

Reviewer #1: Yes

Reviewer #2: Yes

3. Have the authors made all data underlying the findings in their manuscript fully available (please refer to the Data Availability Statement at the start of the manuscript PDF file)?

Reviewer #1: Yes

Reviewer #2: Yes

4. Is the manuscript presented in an intelligible fashion and written in standard English?

Reviewer #1: Yes

Reviewer #2: Yes

5. Review Comments to the Author

Reviewer #1: The present population-based study has scientific merit because analyzed mortality data from Brazil's mortality system, covering

the period from 2015 to 2022. I believe that the results are interesting, but perhaps they can be applied to the local epidemiological reality. I have made some recommendations below.

- Case detection using ICD-10 may be biased. How did the authors correct these possible biases?

- How were the outcomes measured? Perhaps ICHOM is suitable for this study.

- Regarding continuous variables, have their distributions been evaluated? The Kolmogorov-Smirnov test with Lilliefors correction was applied;

- There are few variables presented, I suggest showing more sociodemographic and clinical data of the sample evaluated.

- In some situations in table 1, the 95% CI has the same values for the lower and upper limits. How is this possible?

- A survival curve using the Kaplan-Meyer method could have been performed.

- Calculate the statistical power of the evaluated sample.

- Are cost variables available? The case-mix could be presented and added to the multivariate model.

- Clinical variables related to oxygen therapy could be shown and perhaps added to the multivariate model.

- What are the statistical criteria used to use Poisson regression?

Reviewer #2: I read with appreciation the interesting PGPH-D-23-02038 manuscript intitled “Persistent high mortality rates for Diabetes Mellitus and Hypertension after excluding deaths associated with COVID-19 in Brazil, 2020-2022” and some questions emerged.

To a full paper the introduction section is very short and discuss modestly the diseases evaluated and its epidemiological role in terms morbidity and mortality impact worldwide and in Brazil, before and during pandemic period. In other words, the study rationale seems absent and also there is no a hypothesis or something like.

The sentence “even though the role of HTN and DM as independent risk factors are not yet clear (8)” needs a robust support, specifically regarding DM.

The excerpt "In this study we compared sex, age and state of residence adjusted mortality ratios (aMR) in Brazil in 2020-2022, compared to the preceding period of 2015-2019 and in subgroups of CVD, DM and HTN whenever these conditions were mentioned on death certificates. Comparisons were made with and without COVID-19 mentioned on the death certificates.", seems more appropriate into methods section, because the authors explain analytical procedures instead to state the overall manuscript purpose, for example. The third paragraph, could be better exploited, highlighting the CVD, HTM and DM rates or data in Brazil apart from a brief message to the reader on the indirect pandemic effects, including those taken as residual.

Please, to update the dataset source because the link https://opendatasus.saude.gov.br/dataset/sim is broken.

This is the latest version available to the 2022 year? There were exclusion criteria applied?

Please, to clarify what is the reason for differences between values to the total deaths added in the present analysis (11,423,288) regarding the total deaths available in “TabNet/Datasus” from Brazilian Ministry of Health (11,484,763 deaths).

The excerpt “All data from death certificates except information that could identify individuals were available for analysis…” needs more attention since the Brazilian Ministry of Health does not offer “All data from death certificates” but rather a simplified digital version, including variables solely administrative.

Regarding sentence “…we worked with ICD-10 codes of interest mentioned in any field from the death certificates” the authors have worked with ICD-10 codes in “any field” or specific fields such as 49 field (section VI)?

The Figure 1 might be improved through the transformation in a panel with four figures (a, b, c and d), where de first would be a representation of the overall mortality rate (axis Y) and its respective confidence interval for each year (axis X). The second, third and fourth figures could follow the same logic. The Table 2 is short of objectivity.

Regarding on limitations study, the authors should explain to the reader the expected impact of the underreporting and misclassification errors on the interpretation of main manuscript results. It would also be interesting to point out the unavailability of census 2022 data for Brazilian population and the limits of the projected population by the IBGE during 2015 to December 2022.

6. PLOS authors have the option to publish the peer review history of their article (what does this mean?). If published, this will include your full peer review and any attached files.

**Do you want your identity to be public for this peer review?** For information about this choice, including consent withdrawal, please see our Privacy Policy.

Reviewer #1: No

Reviewer #2: **Yes: **Jesem Orellana

---

## [Editor Report · Decision Letter 1]

26 Mar 2024

Persistent high mortality rates for Diabetes Mellitus and Hypertension after excluding deaths associated with COVID-19 in Brazil, 2020-2022

PGPH-D-23-02038R1

Dear Dr. Rodrigo Moreira,

We are pleased to inform you that your manuscript 'Persistent high mortality rates for Diabetes Mellitus and Hypertension after excluding deaths associated with COVID-19 in Brazil, 2020-2022' has been provisionally accepted for publication in PLOS Global Public Health.

Best regards,

Jonas Wolf

Guest Editor

The authors performed all of the reviewers' suggestions. The manuscript may be accepted for publication.